# Liver Disease Screening and Hepatitis C Virus Elimination in Taiwan Rural Indigenous Townships: Village-By-Village Screening and Linking to Outreach Hepatology Care

**DOI:** 10.3390/ijerph19063269

**Published:** 2022-03-10

**Authors:** Hui-Min Tien, Tai-Chung Cheng, Hsiao-Chu Lien, Kuei-Fei Yang, Cherng-Gueih Shy, Yu-Ling Chen, Nien-Tzu Hsu, Sheng-Nan Lu, Jing-Houng Wang

**Affiliations:** 1Laiyi Primary Health Center, Pingtung County 922, Taiwan; zsazsatien@yahoo.com.tw (H.-M.T.); keejwe@gmail.com (H.-C.L.); 2Mudan Primary Health Center, Pingtung County 945, Taiwan; cttcttcttctt@yahoo.com.tw (T.-C.C.); vais1027@gmail.com (K.-F.Y.); 3Public Health Bureau, Pingtung County 900, Taiwan; graycgshy@gmail.com; 4Division of Hepato-Gastroenterology, Department of Internal Medicine, Kaohsiung Chang Gung Memorial Hospital and Chang Gung University College of Medicine, Kaohsiung City 833, Taiwan; lillian5051222@gmail.com (Y.-L.C.); e19911221@gmail.com (N.-T.H.); juten@ms17.hinet.net (S.-N.L.); 5Kaohsiung Research Association for the Control of the Liver Diseases, Kaohsiung City 833, Taiwan

**Keywords:** hepatitis B surface antigen, antibody to hepatitis C virus, gamma-glutamyl transferase, indigenous peoples, community screening, alcohol consumption

## Abstract

Medical resources are limited for hepatitis C virus (HCV) elimination in rural indigenous areas of Taiwan. This study aimed to investigate liver disease risk and conduct a HCV elimination program in two rural indigenous townships. A program of village-by-village screening tests was conducted including hepatitis B virus surface antigen (HBsAg), antibody to HCV (anti-HCV) and gamma-glutamyl transferase (GGT), linking to outreach hepatology care at two indigenous townships (Laiyi and Mudan). Adult residents were invited to join this program. One hepatology specialist assessed liver disease risk, provided HCV treatment counselling and initiated direct acting antivirals (DAA) at an outreach hepatology clinic in primary health centers. A total of 3503 residents attended this program with a screening coverage of 73.5%. The prevalence of HBsAg, anti-HCV, and high GGT level was 8.2%, 10.0% and 19.5%, respectively. Laiyi had significantly higher prevalence of anti-HCV than Mudan. While males had significantly higher prevalence of HBsAg and high GGT in both townships, females in Laiyi had higher anti-HCV prevalence. HBsAg and high GGT prevalence peaked at 40–59 years of age and anti-HCV prevalence increased significantly with age. Two hundred and sixty-three residents visited the outreach hepatology clinic for HCV treatment evaluation, with 121 (46%) residents having active HCV, while 116 received DAA, with 111 (95.7%) achieving HCV elimination. For rural indigenous townships in southern Taiwan, HCV infection and alcohol consumption were two major liver disease risks. While HCV infection was predominant in old females, chronic hepatitis B virus infection and habitual alcohol consumptions predominated in middle-aged males. HCV elimination was achieved by the village-by-village screening model and linked to outreach hepatology care.

## 1. Introduction

Chronic viral hepatitis, including hepatitis B virus (HBV) and C virus (HCV) infection, along with its complications, is a common cause of liver-related mortalities in Taiwan. Chronic HCV infection can progress to cirrhosis and hepatocellular carcinoma and lead to increased mortality from hepatic and extrahepatic disease in various communities [1,2]. The global prevalence of active HCV was estimated to be 1% in a modeling study in 2015, with substantial increased prevalence in special groups including persons who inject drugs, men having sex with men, and patients on maintenance hemodialysis [3,4,5,6]. In Taiwan, the estimated prevalence of HCV infection was 1.8–5.5% with geographic aggregation; however, the prevalence of first-time blood donors has decreased by 71% in the last two decades [7,8]. While the World Health Organization (WHO) aims to eliminate HCV in 2030, Taiwan aims to achieve this WHO goal by 2025 [9,10]. To achieve this, several challenges including sustainable financing, effective and efficient screening, continuum of care and improving accessibility exist [10]. In Taiwan, township-specific risk maps of liver disease have been sketched based on seven nationwide surrogate markers in 2018 [11], with the map being used as reference to allocate priority for HCV elimination. Indigenous populations are designated as special groups, with dwelling places located in rural and mountain townships.

There are eight rural indigenous townships of Pingtung County localized at the most southern part of Taiwan. Among these townships, Laiyi and Mudan are at high and intermediate risk of liver diseases with high priorities for HCV elimination programs [11], being located in rural mountain areas with limited medical resources. Taiwan is an endemic area of chronic HBV infection. In addition to chronic HBV and HCV infection, high prevalence of alcoholism and alcoholic-related liver disease is also associated with high risk of mortality in indigenous populations [12,13]. Screening and linking to accessible care form a model of HCV elimination in rural areas. For HCV treatment, outreach hepatology specialist care might increase treatment uptake for indigenous people. Therefore, a program for HCV elimination in these two rural aboriginal townships was proposed, including village-by-village screening and linking to outreach hepatology specialist care. The aims of this project were to elucidate the etiology of liver risk, and to conduct an elimination program for HCV infection in these two rural aboriginal communities. 

## 2. Materials and Methods

### 2.1. Study Sites and Participants

Laiyi and Mudan are two rural indigenous townships with a combined population of 12,268 registered residents in 2020 (Figure 1; our own map). People living in these two townships for at least 4 days a week were defined as frequent residents. Altogether, there are seven and six villages in Laiyi and Mudan townships, respectively, all with limited medical resources. Each township has two general practitioners providing clinical and public health services in the primary health center in the largest villages and outreaching to primary health stations in almost every other village. The study population was obtained in January 2020, by age, gender and village, from the website of the local household registration office, while the existing database from previous screenings and clinical medical records of the primary care center was reviewed. This study was approved by the Institutional Review Board of Chang Gung Memorial Hospital.

The two rural indigenous townships of Laiyi and Mudan are located in Pingtung County, Taiwan. There are seven and six villages in Laiyi and Mudan townships respectively.

### 2.2. Community-Based Screening and Links to Accessible Care

All residents aged 30 years or older in Laiyi and Mudan were invited to attend the village-by-village screening tests for liver diseases, including hepatitis B virus surface antigen (HBsAg), antibody to HCV (anti-HCV) and gamma-glutamyl transferase (GGT) as a surrogate marker of habitual alcohol drinking. In addition, the existing database of HBsAg and anti-HCV were reviewed from previous screening programs and the clinical medical records of the primary health center. Residents with positive anti-HCV were appointed to the special clinic in the primary health center or stations for further evaluation of active HCV infection, including quantitative HCV-RNA (Roche TaqMan HCV assay, Roche Molecular Systems, Inc., Branchburg, NJ, USA). A hepatology specialist from a medical center made the diagnosis, performed liver ultrasonography, assessed liver disease severity, and provided HCV treatment counselling in the outreach clinic at primary health centers or stations. For those residents eligible for HCV treatment, direct acting antivirals (DAA) were initiated according to the guidelines of National Health Insurance for treatment or reimbursement.

### 2.3. Statistics

Quantitative variables were expressed with mean ± standard deviation (SD) or median within an established range; qualitative variables were expressed as absolute and relative frequencies, while gender difference, age trend and village discrepancy were analyzed. To calculate the coverage rate of hepatitis screening, results of this community-based screening and the existing database from previous screenings and the clinical medical records of the primary care center were merged with the estimated coverage rate, adjusted by proportion of frequent residents. The calculated equation was (total participants of screening/total population)/proportion of frequent residents. While the Student’s t-test was used for comparisons of quantitative variables, Chi-square and Fisher’s exact test were used in categorical variables; the Cochran-Armitage trend test was used to test the trend in age-specific prevalence, and geographical distribution and prevalence of anti-HCV were performed using QGIS 3.6 software. Statistical analysis was performed using SAS version 9.4 (SAS Institute, Cary, NC, USA). All *p* values were derived from 2-tailed tests, and a level of <0.05 was accepted as statistically significant.

## 3. Results

### 3.1. Resident and Screen Coverage Rate

Between Jan. 2020 and Apr. 2021, a total of 8372 (*n* = 4974 in Laiyi, and *n* = 3398 in Mudan) residents was invited to attend this program by contact with public health nurses or by telephone call. There were 3503 (41.8%, mean age: 59.7) residents attending the screening program, including l784 (35.9%, mean age: 59.6) in Laiyi township and 1719 (50.6%, mean age: 59.9) in Mudan township. By field investigation, 52.0% and 64.1% of the population were frequent residents in Laiyi and Mudan, respectively. After adjusting the proportion of frequent residence, the overall screen coverage rate was 73.5%, with 69.0% in Laiyi and 78.9% in Mudan (Table 1).

### 3.2. Prevalence of HBsAg, Anti-HCV and High GGT Level

The prevalence of HBsAg, anti-HCV, and high GGT (defined as more than 2 × upper normal limit) were 8.2%, 10.0% and 19.5% (7.5%, 15.8%, 12.7% in Laiyi, and 9.0%, 4.1%, 24.3% in Mudan) (Table 1). While there was no significant difference in the prevalence of HBsAg, Laiyi township had a significantly higher prevalence of anti-HCV (*p* < 0.001) and lower prevalence of high GGT (*p* < 0.001) compared with Mudan township. In both townships, there was significantly higher prevalence of HBsAg (Laiyi: *p* = 0.001, Mudan: *p* = 0.011) and high GGT (Laiyi: *p* < 0.001, Mudan: *p* < 0.005) in males, more than in females (Table 2). While there was no significant difference in gender in Mudan township, the prevalence of anti-HCV was higher in females than males in Laiyi township (Laiyi 17.7% vs. 12.8%, *p* = 0.005) (Table 2). There were significant inter-village differences in the prevalence of HBsAg and anti-HCV (Table 2). The age-specific prevalence of HBsAg and high GGT showed inverted U-shapes with a peak at 40–59 years for both townships (Figure 2). The prevalence of anti-HCV increased by age (*p* for trends, *p* < 0.001) and the difference between the two townships also increased by age (*p* for trends, *p* < 0.001).

### 3.3. HCV Treatment

Among the 351 residents with anti-HCV, 263 (74.9%) visited an outreach hepatology clinic, including 201 in Laiyi and 62 in Mudan. HCV viremia was detected in 121 (46%) residents including 95 and 26 in Laiyi and Muden, respectively. One hundred and sixteen residents received DAA treatments, including 93 in Laiyi and 23 in Mudan, while five residents without DAA treatments included two alcoholic residents with poor compliance, one with cancer under active treatment, one with serious drug–drug interaction with DAA, and one moving to another city. HCV elimination with sustained virological response (SVR), defined as undetectable HCV viremia for at least 12 weeks after end of treatment, was obtained in 111 (95.7%) residents (Table 3).

## 4. Discussion

While some hyperendemic areas of HCV infection have been located in southern Taiwan [7], this study showed high prevalence and geographical variation among and within indigenous townships. Using a model of community-based screening and linking to hepatology care, this study also demonstrated high HCV treatment uptake and SVR in these two rural indigenous townships; additionally, there was high prevalence of high GGT which might reflect high prevalence of habitual alcohol intake, as HBV infection and alcohol drinking were predominant in middle aged men, while HCV infection was predominant in old women.

The prevalence of chronic HBV infection was 13–25%, with geographical variations in Taiwan [14]. Although a universal HBV vaccination program has been implemented in Taiwan since 1984, most adult residents in this study were born before this program and without HBV vaccination. Indigenous groups in southern Taiwan had a HBsAg-positive rate of 22–25% in a previous report [15]; however, the prevalence of HBsAg-positive was 7.5–9.0% in this study. While the previous report was a cross-sectional study conducted in 1988 [15], this study was a well-designed community screening program conducted in 2020 with one of its aims an investigation of liver disease etiology. Furthermore, another report demonstrated a larger proportion of HBV genotype B in indigenous groups than the general population in Taiwan, which might result in higher spontaneous HBsAg seroconversion and lower HBsAg-positive rate in indigenous areas [16,17]. This study has accordingly provided more recent and informative HBV epidemiology concerning indigenous areas of southern Taiwan.

For areas with HCV prevalence of more than 2–5%, general population screening has been recommended by health guidelines [18]. In large-scale free hepatitis screening in Taiwan, HCV seroprevalence was estimated to be 4.4% with variations among counties [14], although geographical variations existed not only between townships but also between villages within the county [19]. The prevalence of HCV in Taiwan was 14.2–35.1% for adults in indigenous areas, increasing with age [20,21,22]. The attributable risk of HCV infection was mostly related to non-sterile injections before the 1980s [8,19]. This study demonstrated variations of HCV seroprevalence between and within indigenous townships, which is compatible with previous reports and the township-specific risk maps for liver disease from the Ministry of Health and Welfare in Taiwan [11,22]. 

General adult HCV screening with a high screening coverage rate might be necessary to achieve the goal of identifying 90% of all HCV infections set by the WHO [9]. To meet this goal, the screening strategies could be tailored according to local epidemiology of HCV infection [10]. Community studies have demonstrated that outreach continuous and small-scale screening programs increased accessibility and cost-effectiveness in community HCV screening [23,24]. As a result, this program conducted village-by-village screening to improve the coverage and cost-effectiveness of screening. Although this screening program had a high coverage rate of 73.5%, this still might not be enough to meet the WHO target, and could be continued with ongoing screening to increase screening coverage to diagnose 90% of all HCV infections in these rural indigenous areas. Further measures might be necessary however, to scale-up the screening coverage, including enhancing awareness of HCV infection and increasing media coverage of news related to HCV treatment [25].

The implementation of high efficacious and low side-effect DAA for HCV treatment has made HCV elimination feasible, but multiple and structural barriers remain to be overcome, including accessibility of professional care [26]. Although primary-care-based services have increased HCV treatment uptake and obtained similar outcomes to those of specialty clinics for uncomplicated patients, professional care might be necessary for more complicated cases [27,28], and, additionally, local regulations have hindered the model of HCV treatment by primary care physicians in Taiwan, so this study provided outreach specialty care at primary health centers and stations to increase HCV treatment uptake. 

Results have demonstrated that DAA treatment uptake and SVR captured more than 95% of those cases with active HCV with good treatment compliance. Similar to interferon-based therapy, DAA might have similar efficacy between indigenous populations and other groups [29]. This model of village-by-village screening and linking to outreach hepatology care was effective in achieving the WHO 80% treatment coverage rate for rural indigenous townships; however, further efforts are required to increase the treatment uptake and compliance for residents with habitual alcohol consumption.

For indigenous people, high prevalence of unhealthy drinking leads to alcohol-related disorders that appear to increase mortality and a health gap between indigenous populations and the general population [12,13,30], as habitual alcohol drinking has been found to increase the risk of chronic liver disease for HCV infection among indigenous peoples in Taiwan [31,32]. The conventional GGT test, a marker of the oxidative stress associated with ethanol metabolism, is elevated in regular drinkers with a long drinking history [33]. Although non-specific for alcohol use, the GGT test is widely available and inexpensive for community screening, so was accordingly performed to detect habitual alcohol drinking in this screening program. To increase the specificity, habitual alcohol drinking was defined as two-fold elevation of GGT level. However, structured and validated screening tools including the alcohol use disorders inventory test, specific alcohol biomarkers, physical exam and clinical interview will be necessary to confirm harmful alcohol use and alcohol liver disease [34]. 

The prevalence of high GGT was 19.5% with variation between two townships, but did not reveal significant variation among villages within each townships. Compared with the prevalence of 40–55% alcoholism by psychiatric interview 25 years earlier [12,31], this study result showed a lower prevalence, which might be explained by difference in detection methods and study cohorts, and although this prevalence might be underestimated, habitual drinking is still one major liver disease risk that needs to be improved in these two rural indigenous townships.

## 5. Conclusions

HCV infection and habitual alcohol drinking were the two major risks for liver diseases in rural indigenous townships in southern Taiwan, an endemic area of chronic HBV infection. Geographical variations among and within indigenous townships existed, for HBV and HCV infections. While HCV infection was predominant in older females, chronic HBV infection and habitual alcohol consumption were prevalent in middle-aged males. HCV elimination could be achieved by the model of village-by-village screening linked to outreach hepatology care.

## Figures and Tables

**Figure 1 ijerph-19-03269-f001:**
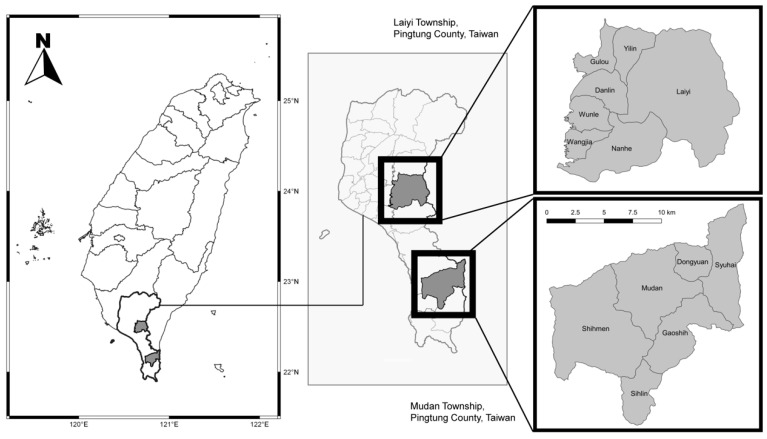
The locations of the two rural indigenous townships (our own map).

**Figure 2 ijerph-19-03269-f002:**
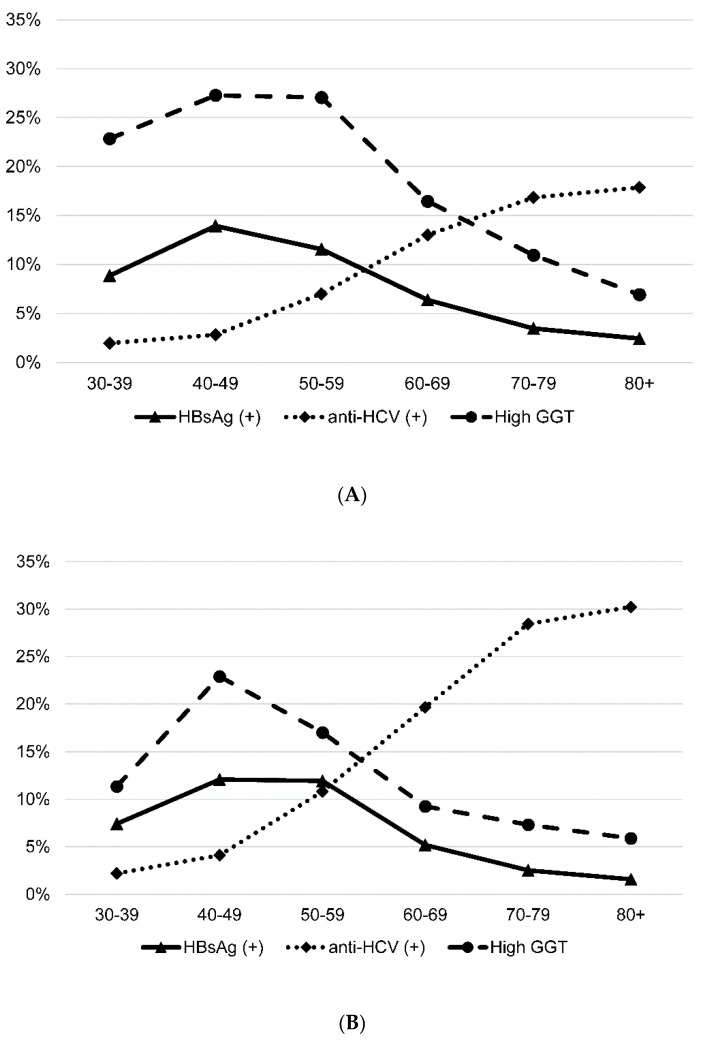
Age-specific prevalence of hepatitis B surface antigen (HBsAg), antibody to hepatitis C virus (anti-HCV) and high gamma-glutamyl transferase (GGT). (**A**) Total screening of residents in both communities, (**B**) Laiyi township and (**C**) Mudan township.

**Table 1 ijerph-19-03269-t001:** Demographics, screen coverage rate and prevalence of hepatitis B virus surface antigen (HBsAg), antibody to hepatitis C virus (anti-HCV) and high gamma-glutamyl transferase (GGT).

	Total (*n* = 3503)	Laiyi (*n* = 1784)	Mudan (*n* = 1719)
Age (year, mean ± SD)	59.7 ± 13.7	59.6 ± 13.1	59.9 ± 14.3
Sex (%)			
Male	1445 (41.3)	719 (40.3)	726 (42.2)
Female	2058 (58.7)	1065 (59.7)	993 (57.8)
Screen coverage rate (%)	41.8	35.9	50.6
Adjusted screen coverage rate (%)	73.5	69.0 ^#^	78.9 ^##^
Positive HBsAg (%)	8.2	131/1739 (7.5) *	154/1719 (9.0) *
Positive Anti-HCV (%)	10.0	281/1784 (15.8) **	70/1717 (4.1) **
Hight GGT (%)	19.5	114/900 (12.7) ***	316/1302 (24.3) ***

High GGT: gamma-glutamyl transferase (defined as ≥2 × upper normal limit). ^#^: Laiyi frequent residence 52.0%; ^##^: Muden frequent residence 64.1%. *: *p* = 0.128; **: *p* < 0.001; ***: *p* < 0.0001.

**Table 2 ijerph-19-03269-t002:** Screen coverage rate and prevalence of hepatitis B virus surface antigen (HBsAg), antibody to hepatitis C virus (anti-HCV) and high gamma-glutamyl transferase (GGT) stratified by gender and village.

	Coverage Rate	HBsAg (+)	Anti-HCV(+)	High GGT
Laiyi township				
Gender-specific (%)				
Male	719/2467 (29.1)	70/697 (10.0)	92/719 (12.8)	92/393 (23.4)
Female	1065/2507 (42.5)	61/1042 (5.9)	189/1065 (17.7)	22/507 (4.3)
*p*-value	<0.001	0.001	0.005	<0.001
Village-specific (%)				
Danlin	234/577 (40.6)	17/222 (7.7)	68/234 (29.1)	17/123 (13.8)
Wunle	228/677 (33.7)	12/224 (5.4)	15/228 (6.6)	8/103 (7.8)
Gulou	298/885 (33.7)	26/279 (9.3)	83/298 (27.9)	24/168 (14.3)
Laiyi	271/737 (36.8)	22/268 (8.2)	62/271 (22.9)	15/131 (11.5)
Nanhe	364/945 (38.5)	40/363 (11.0)	12/364 (3.3)	25/152 (16.4)
Wangjia	200/689 (29.0)	5/199 (2.5)	7/200 (3.5)	11/117 (9.4)
Yilin	189/464 (40.7)	9/184 (4.9)	34/189 (18.0)	14/106 (13.2)
*p*-value	<0.001	0.005	<0.001	0.415
Mudan township				
Gender-specific (%)				
Male	726/1687 (43.0)	80/726 (11.0)	31/726 (4.3)	176/546 (32.2)
Female	993/1711 (58.0)	74/993 (7.5)	39/991 (3.9)	140/756 (18.5)
*p*-value	<0.001	0.011	0.729	<0.001
Village-specific (%)				
Sihlin	214/463 (46.2)	12/214 (5.6)	12/214 (5.6)	55/175 (31.4)
Shihmen	665/1279 (52.0)	66/665 (9.9)	23/664 (3.5)	93/445 (20.9)
Syuhai	168/317 (53.0)	17/168 (10.1)	7/168 (4.2)	39/141 (27.7)
Mudan	290/543 (53.4)	29/290 (10.0)	7/289 (2.4)	61/245 (24.9)
Dongyuan	174/355 (49.0)	22/174 (12.6)	17/174 (9.8)	41/157 (26.1)
Gaoshih	208/441 (47.2)	8/208 (3.8)	4/208 (1.9)	27/139 (19.4)
*p*-value	0.093	0.016	0.001	0.059

High GGT: gamma-glutamyl transferase (defined as ≥2 × upper normal limit).

**Table 3 ijerph-19-03269-t003:** Demographics of hepatitis C viremia patients and effect of direct acting antivirals (DAA) in outreach hepatology clinic.

	Total (*n* = 121)	Laiyi (*n* = 95)	Mudan (*n* = 26)
Age (year, mean ± SD)	69.1 ± 11.6	69.6 ± 11.3	67.2 ± 12.6
Sex (%)			
Male	44 (36.4)	31 (32.6)	13 (50)
Female	77 (63.6)	64 (67.4)	13 (50)
DAA ^#^ (%)	116 (95.9)	93 (97.9)	23 (88.5)
SVR ^##^ (%)			
IT	111 (91.7)	88 (92.6)	23 (88.5)
PP	111 (95.7)	88 (94.6)	23 (100)

SVR: sustained virological response defined as undetectable HCV viremia at least 12 weeks after treatment; IT: intension-to-treat analysis; PP: per-protocol analysis. ^#^: Five without DAA due to alcoholism with poor compliance, cancer under active treatment, serious drug–drug interaction, and residence movement. ^##^: Five without SVR: one died during treatment and four had poor compliance.

## Data Availability

The datasets are available from the corresponding author with reasonable request.

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
