# Peer review of "Liver Disease Screening and Hepatitis C Virus Elimination in Taiwan Rural Indigenous Townships: Village-By-Village Screening and Linking to Outreach Hepatology Care"

_ijerph, 2022, doi:10.3390/ijerph19063269_

Round 1

Reviewer 1 Report

Results should be better discussed, starting from the data in the Tables and Graphs presented. 
Discussions by age, sex and the two villages for HBV and HCV infections should be improved.
Conclusions must be rewritten both in the Abstract and at the end of the paper, starting from all the results, graphs and the statistics in the end.

Author Response

  1. Results should be better discussed, starting from the data in the Tables and Graphs presented. 

Response: Thanks for the reviewer’s comments.

  1. Discussions by age, sex and the two villages for HBV and HCV infections should be improved.

Response: This is a program for HCV elimination in rural indigenous areas.  Before this program, we conducted community screening to find the patients with active HCV.

  1. Conclusions must be rewritten both in the Abstract and at the end of the paper, starting from all the results, graphs and the statistics in the end.

Response: We modified the conclusions in the abstract and end of paper according to the comments.  

Reviewer 2 Report

Dear authors,

Congratulations to the topic of this study.

The only details to refine the english is adjust the word peoples, line 237.

There are words without space in line 192 .

Detailed comments:

The study is very interesting describing the Program to eliminate Hepatitis C, but there are a lot of results that could be more explained.

  1. Modification the title since the article did not work only with  Hepatitis C and its eliminations, this modification could be associated to keywords.
  2. The first paragraph of introduction could be generic including, Hepatitis B and others.
  3. The methods is generic and it is not a problema but, I need a answer what is the procedure with  HBS positive patients? I see molecular methods to Hepatitis C.  
  4. Results there are coinfections? What is the percentage.
  5. Hepatitis B the authors could discuss this in relation to Delta Hepatitis topic.
  6. Rewrite the discussion and conclusion.  The article is very interesting top put a evidencie point the politic to elimination of Hepatitis C virus.

Author Response

Congratulations to the topic of this study.

  1. The only details to refine the english is adjust the word peoples, line 237.

Response: The word “peoples” has been changed to “people”.

  1. There are words without space in line 192 .

Response: The form was checked.

Detailed comments:

The study is very interesting describing the Program to eliminate Hepatitis C, but there are a lot of results that could be more explained.

  1. Modification the title since the article did not work only with Hepatitis C and its eliminations, this modification could be associated to keywords.

Response: The title had been modified to “Liver disease screening and hepatitis C virus elimination in Taiwan rural indigenous townships: village-by-village screening and linking to outreach hepatology care” according to the comment.

  1. The first paragraph of introduction could be generic including, Hepatitis B and others.

Response: We had modified the introduction section to include hepatitis B virus description.

  1. The methods is generic and it is not a problem but, I need a answer what is the procedure with  HBS positive patients? I see molecular methods to Hepatitis C.  

Response: Those residents with positive HBsAg were also called back to outreach hepatology clinic. In addition to liver sonography for hepatoma screening, HBV-DNA was also checked. For those patients eligible for antiviral treatment, oral nucleos(t)ides were prescribed according to reimbursement policy and guidelines of national health insurance. Since this is a HCV elimination program, HBV management was not detailed in this manuscript.

  1. Results there are coinfections? What is the percentage.

Response: There are eleven patients (0.3%) with coinfected HBV and HCV.

  1. Hepatitis B the authors could discuss this in relation to Delta Hepatitis topic.

Response: We did not check antibody to hepatitis delta virus or HDV-RNA for those residents with positive HBsAg. Therefore, we did not discuss this topic. However, the prevalence of HDV was low (1.15%) in Taiwan. The coexisting HDV infection did not influence the clinical manifestation of patients with chronic HBV infection in study (Lee WC, et al. Investigating the prevalence and clinical effects of hepatitis delta viral infection in Taiwan. J Microbiol Immunol Infect 2021;54:901-08)

  1. Rewrite the discussion and conclusion.  The article is very interesting top put a evidence point the politic to elimination of Hepatitis C virus.

Response: We had modified the sections of discussion and conclusion.

Reviewer 3 Report

This is an interesting paper.  From an English language perspective, although some of the prose is a bit awkward (but still acceptable) there are some minor errors (e.g. "man" instead of "men" etc) and I would ask the authors to review from that perspective (ie. English proficiency).

My specific comments are as follows:

  1. GGT is very non-specific and the use of the GGT as a marker of alcohol consumption is no longer considered acceptable in North America.  It can be a marker of almost any liver injury/disease including non-alcoholic fatty liver disease.  The authors should have included a full panel of liver biochemistry (ie. ast, alt, alkphos, and ggt).  The authors should either remove the GGT sections from the manuscript or provide a more thorough and critical commentary in the Discussion about its non-specificity.  
  2. The seroprevelance of HCV in this Indigenous community is high by North American standards and there needs to be a greater commentary in the Discussion about why the prevalence is so high in the Taiwanese Indigenous community (ie. due to endemic injection drug use, blood transfusions, exchange of body fluids through regional cultural practices?).  A North American audience would be very interested in an explanation as to the high prevalence.
  3. The prevalence of HBsAg carriage is also high.  Taiwan is reported to have undertaken a program of universal HBV vaccination on a population basis in the early 1980s and therefore the high prevalence is surprising.  This requires a commentary in the Discussion.  Did this Indigenous community not receive vaccination?  Was the vaccination program unevenly applied to the population?

Author Response

This is an interesting paper.  From an English language perspective, although some of the prose is a bit awkward (but still acceptable) there are some minor errors (e.g. "man" instead of "men" etc) and I would ask the authors to review from that perspective (ie. English proficiency).

Response: Thanks for review’s comments. We had corrected the errors. The manuscript has been edited by native English-speaking teachers before submission.

My specific comments are as follows:

  1. GGT is very non-specific and the use of the GGT as a marker of alcohol consumption is no longer considered acceptable in North America.  It can be a marker of almost any liver injury/disease including non-alcoholic fatty liver disease.  The authors should have included a full panel of liver biochemistry (ie. ast, alt, alkphos, and ggt).  The authors should either remove the GGT sections from the manuscript or provide a more thorough and critical commentary in the Discussion about its non-specificity.  

Response: Thanks for reviewer’s comments. Structured and validated questionnaires is suggested as the screening tool of alcohol use. Specific alcohol biomarkers, including urine and hair ethyl glucuronide, are preferred to aid the diagnosis of alcohol use. High cost of these tests and limited availability might hinder the use of these specific tests. Despite non-specific, elevated GGT level might suggest alcohol use in indigenous areas where the prevalence of habitual alcohol use was high in previous reports. In the discussion section, we mentioned the non-specificity of GGT in the diagnosis of alcohol use and cited a paper [reference 34] for review of this issue.

  1. The seroprevelance of HCV in this Indigenous community is high by North American standards and there needs to be a greater commentary in the Discussion about why the prevalence is so high in the Taiwanese Indigenous community (ie. due to endemic injection drug use, blood transfusions, exchange of body fluids through regional cultural practices?).  A North American audience would be very interested in an explanation as to the high prevalence.

Response: The attributable risk of high HCV infection might be related to unsterile medical injection before 1980s when non-disposable syringes or needles were commonly used in Taiwan. We explained it in the discussion section and cited two references (reference 8 and 19).

  1. The prevalence of HBsAg carriage is also high.  Taiwan is reported to have undertaken a program of universal HBV vaccination on a population basis in the early 1980s and therefore the high prevalence is surprising.  This requires a commentary in the Discussion.  Did this Indigenous community not receive vaccination?  Was the vaccination program unevenly applied to the population?

Response: Universal HBV vaccination program has been implemented in Taiwan since 1984. Most adult residents in this study are the birth cohort before universal HBV vaccination. We explained this point in the discussion section.